# Genetic Characteristics of Canine Adenovirus Type 2 Detected in Wild Raccoon Dogs (*Nyctereutes procyonoides*) in Korea (2017–2020)

**DOI:** 10.3390/vetsci9110591

**Published:** 2022-10-27

**Authors:** Yoon-Ji Kim, Sook-Young Lee, Young-Sik Kim, Eun-Jee Na, Jun-Soo Park, Jae-Ku Oem

**Affiliations:** 1Laboratory of Veterinary Infectious Disease, College of Veterinary Medicine, Jeonbuk National University, Iksan 54596, Korea; 2Center for Study of Emerging and Re-Emerging Viruses, Korea Virus Research Institute, Institute for Basic Science (IBS), Daejeon 34123, Korea

**Keywords:** *adenovirus*, canine adenovirus type 2, infectious bronchial disease, wildlife

## Abstract

**Simple Summary:**

Adenovirus infection in animals occurs worldwide, and effective protection for pets is achieved through vaccination. Adenovirus vaccination in wild animals is rare, except in the case of rescue, treatment, and return to the wild. This study aimed to investigate the most prevalent canine adenovirus strains in Korean wild raccoon dogs, sequence the entire viral genome, and analyze the genetic characteristics in comparison with the adenovirus strains prevalent worldwide. We isolated one canine adenovirus type 2 (CAdV-2) and sequenced and analyzed its entire gene (CAdV-2/18Ra-54). This strain was found to have a high similarity with the Toronto A26/61 strain, which is widely used in vaccines, suggesting that pet vaccination can cause disease transmission and infection in wild animals. Therefore, further studies on the safety of the vaccine are needed. These studies monitor the epidemiology of CAdV-2 infection and contribute to the understanding of the genetic evolution of CAdV-2 in Korea and the development of new vaccines.

**Abstract:**

Adenovirus has been detected in a wide range of hosts like dogs, foxes, horses, bats, avian animals, and raccoon dogs. Canine adenoviruses with two serotypes host mammals and are members of the mastadenovirus family. Canine adenovirus type 1 (CAdV-1) and canine adenovirus type 2 (CAdV-2) cause infectious canine hepatitis and infectious bronchial disease, respectively. In this study, we investigated the prevalence of CAdV-1 and 2 in wild *Nyctereutes procyonoides* in Korea in 2017–2020 from 414 tissue samples, including the liver, kidney, lung, and intestine, collected from 105 raccoon dog carcasses. Only CAdV-2 was detected in two raccoon dogs, whereas CAdV-1 was not detected. Tissue samples from raccoon dogs were screened for CAdV-1 and CAdV-2 using conventional PCR. Adenovirus was successfully isolated from PCR positive samples using the Vero cell line, and the full-length gene sequence of the isolated viruses was obtained through 5’ and 3’ rapid amplification of cDNA ends (RACE). The major genes of the isolated CAdV-2/18Ra54 and CAdV-2/18Ra-65 strains showed the closest relationship with that of the CAdV-2 Toronto A26/61 strain isolated from Canada in 1976. There is no large mutation between CAdV-2, which is prevalent worldwide, and CAdV-2, which is prevalent in wild animals in Korea. In addition, it is still spreading and causing infections. The Toronto A26/61 strain, which showed the most similarity to CAdV-2/18Ra-54, was likely transmitted to wild animals through vaccinated companion animals, suggesting that further research is needed on safety measures surrounding animal vaccination. This study provides information on the genetic characteristics and prevalence of canine adenovirus in domestic wild animals and provides a better understanding of canine adenovirus.

## 1. Introduction

Canine adenoviruses are medium-sized (approximately 90–100 nm) non-enveloped double-stranded DNA viruses (32 kb) with icosahedral symmetry [1,2]. Structurally, adenoviruses are composed of capsid, minor, and core proteins. Capsid proteins are structural proteins and contain fiber, hexon, and penton bases. Among these capsid proteins, fibers bind the adenovirus to the host cell. Hexon is a major capsid protein and is an important target for the genetic modification of adenoviral vectors [3]. Penton base (penton) is a general adenovirus antigen that induces complement-binding antibodies at the fiber-binding site of the host receptor, and plays an important role in infection by inducing mainly serotype-specific neutralizing antibodies [4]. The core proteins consist of genomic DNA, terminal proteins, and core proteins. The DNA polymerase (*pol*) gene plays an important role in DNA replication. Therefore, as major genes for the analysis of adenovirus, pol, a core protein that plays an important role in DNA replication, and fiber, hexon, and penton genes belonging to the capsid protein, were selected [5,6].

Adenoviruses are classified into five genera, *Atadenovirus*, *Aviadenovirus*, *Ichtadenovirus*, *Siadenovirus*, and *Mastadenovirus*, according to the host species and various host-dependent clinical signs [7]. Mastadenovirus infections, including those of canine adenovirus types 1 and 2, manifest clinical signs in the respiratory and gastrointestinal tracts in mammals. CAdV-1 is known to cause infectious hepatitis, and the clinical symptoms include bladder edema, tonsillitis, and multiple vasculitis [8,9]. CAdV-2 is known as one of the causes of kennel cough—a type of cold—and clinical signs include cough, vomiting, lethargy, high fever, runny nose, and loss of appetite [10,11,12]. However, these characteristic clinical signs are not observed in raccoon dogs [13].

A few studies have reported on the surveillance and prevalence of canine adenovirus in dogs and wildlife in countries such as Japan, Korea, Italy, China, and Turkey; even in the absence of an active epidemic, the virus exists and causes infection in wildlife [14,15,16,17,18]. Wild raccoon dogs are widely distributed, and as one of the representative Korean wild animals, they play a critical role in spreading several viral diseases such as rabies, African swine fever, and parvovirus [19,20,21].

Therefore, in this study, the prevalence of canine adenovirus in domestic and wild raccoons and the genetic characteristics of prevalent canine adenovirus strains in Korea were investigated. In addition, we analyzed the close genetic relationship between canine adenoviruses prevalent in Korea and adenoviruses that occur worldwide.

## 2. Materials and Methods

### 2.1. Samples and Adenovirus Detection

In total, 105 raccoon dog carcasses were legally acquired from the National Institute of Wildlife Disease Control and Prevention. All raccoon dog carcasses were found dead due to road accidents or diseases during 2017–2020. They were stored and transported in freezers before and after necropsy. Information pertaining to the 414 tissue samples (liver, kidney, lung, and intestine) excised from the raccoon dogs is listed in Table 1.

Each raccoon dog tissue sample was crushed in 1 mL of phosphate-buffered saline (PBS) and centrifuged at 3000 rpm for 5 min, after which 200 μL of the supernatant was used for viral DNA extraction. Total DNA was extracted using Wizprep™ Viral DNA/RNA Mini Kit V2 (Wizbiosolution, Seongnam, Korea) according to the manufacturer’s protocol. The infection investigation of CAdV-1 and CAdV-2 was performed using PCR by targeting E3 gene sites of sizes 508 bp and 1030 bp using WizPure ™ FX-PCR 2X Master (Wizbiosolution, Seongnam, Korea) [22,23]. The following primers were used: forward (5′-CGCGCTGAACATTACTACCTTGTC-3′) and reverse (5′-CCTAGAGCACTTCGTGTCCGCTT-3′). The PCR conditions were as follows: initial denaturation at 95 °C for 5 min, followed by 40 continuous cycles of 96 °C for 30 s, 58 °C for 1 min, 72 °C for 1 min, and 5 min incubation at 72 °C. The PCR products were loaded on 1.2% agarose gel, and the PCR positive bands were purified using LaboPass™ Gel Extraction Kit (Cosmogenetec, Daejeon, Korea) and sequenced using Applied Biosystems 3730xl DNA Analyzer (Cosmogenetec, Daejeon, Korea). The sequences were determined through Basic Local Alignment Search Tool (BLAST, version 2.9.0) analysis (http://blast.ncbi.nlm.nih.gov/Blast.cgi, accessed on 27 September 2019).

### 2.2. Entire Genome Sequencing

Gene specific primers (GSP) were prepared for each PCR positive sample, and products with sizes in the range of 329–1091 bp were obtained using a PCR method; the annealing temperature was adjusted between 55–65 °C according to the melting temperature of each primer. The sequence of the entire genome was completed by concatenating the 660–1106 bp-sized products using PCR. Details of the primer sequences used for PCR and sequencing can be found in Appendix A.

The entire genomic sequence of the isolated virus was obtained using 5’ rapid amplification of cDNA ends (RACE), using T4 DNA ligase (TaKaRa, Kusatsu, Shiga, Japan), and 3’ RACE using poly A tail. Through 5’ RACE, a gene sequence of 200 bp from the start of the ITR gene to the position of the 5RACE_GSP2 primer was obtained, and through 3’ RACE, a gene sequence of 700 bp from the position of the 3RACE_GSP1 primer to poly A tail was obtained. The PCR conditions were as follows: initial denaturation at 95 °C for 5 min, followed by 40 continuous cycles of 95 °C for 30 s, 63 °C for 30 s, 72 °C for 30 s, and 5 min incubation at 72 °C. Information on the GSP primer and adapter sequences for 5’ RACE and 3’ RACE can be found in Appendix A.

### 2.3. Adenovirus Isolation

Ground lung tissue supernatants of the PCR positive samples were centrifuged at 3000 rpm for 5 min, filtered through a 0.45 µm filter, and inoculated into Vero cells (African green monkey kidney cells, ATCC CCL-81) (American Type Culture Collection, Manassas, VA, USA) with 80% confluency. The cells were cultured in a 25 cm^2^ cell culture flask using 200 µL of Dulbecco’s Modified Eagle’s Medium (Thermo Fisher Scientific, Waltham, MA, USA) supplemented with 2% fetal bovine serum (Thermo Fisher Scientific, Waltham, Massachusetts, USA) and antibiotic-antimycotic (Thermo Fisher Scientific, Waltham, MA, USA). The infected cells were incubated in a 5% CO_2_ atmosphere at 36 °C and observed twice daily for 10–14 days to confirm the cytopathic effect (CPE). When CPE was evident, it was sub-cultured to the next passage, and the media and cells were collected to confirm virus isolation using PCR. The primer set (E3 gene) used previously for virus detection was employed once again.

### 2.4. Transmission Electron Microscopy

Individual infected cells were observed using a biological Transmission Electron Microscope (bio-TEM). The first fixation was performed overnight at 4 °C using 2% paraformaldehyde and 2% glutaraldehyde in 0.05 M sodium cacodylate buffer, followed by washing thrice with 0.05 M sodium citrate buffer. After fixation, the cells were treated with 1% osmium tetroxide in 0.05 M sodium cacodylate buffer and washed twice with distilled water. En bloc staining with 0.5% uranyl acetate was followed by dehydration using 30–100% ethanol, two transitions using 100% propylene oxide, infiltration using Embled 812 resin, and polymerization at 60 °C for 48 h. The samples were sectioned using ultramicrotome and were placed on the grid. The grid was transferred to 2% uranyl acetate and Reynold’s lead citrate at 7 min intervals, stained, and observed. TEM equipment H-7650 (Hitachi, Tokyo, Japan) was used. All processes except for the one performed at 4 °C were carried out in a fume hood.

### 2.5. Phylogenetic Tree Analysis

Raw sequencing data were aligned and edited using BioEdit version 7.2 with the Clustal W algorithm. The phylogenetic trees were drawn using the neighboring joining (NJ) method using the maximum composite likelihood model with MEGA software version 7. The bootstrap values were calculated with 1000 replicates.

## 3. Results

### 3.1. Detection of Canine Adenoviruses

CAdV-2 was detected in 2 of the 105 raccoon dogs (1.9%), whereas CAdV-1 was not detected. The CAdV-2 strains that were detected were 18Ra-54 and 18Ra-65. Out of 414 tissue samples tested, the presence of CAdV-2 was confirmed as a positive PCR result in six tissues from the liver, kidney, lung, and intestine (Table 2). Both strains were detected in raccoon carcasses that were found in Daejeon in 2018 with no visible clinical signs of canine adenovirus infection other than hair loss. The results of the identification of CAdV-2 18Ra-54 and 65 in each tissue type using PCR are shown in Appendix A.

### 3.2. Virus Isolation

The CPE of canine adenovirus manifests as the phenomenon of the rounding and clustering of cells [24]. Lung tissue infected with 18Ra-54 and 18Ra-65 strains was ground and centrifuged. The supernatant was inoculated into normal Vero cells and observed every day for CPE. Vero cells inoculated with 18Ra-54 strain (CAdV-2/18Ra-54) isolated from the lung tissues showed a rounded, inflated, and clustered presentation at 5 days post-infection (dpi) and became more pronounced after 7 dpi. PCR was performed to confirm the passage number and tissue type, showing a positive result, and CAdV-2/18Ra-54 were confirmed to be found in cells from all passages inoculated with lung tissue from the 18Ra-54 strain. The results are presented in Appendix A. Re-infection of normal Vero cells with CAdV-2/18Ra-54 was confirmed using TEM. Cells presenting CPE at 7 dpi were fixed through physical and chemical pretreatment and observed by TEM. Many adenoviral particles with icosahedral symmetry were observed in the nucleus of infected Vero cells (Figure 1).

In contrast, Vero cells inoculated with the lung tissue of the 18Ra-64 strain did not show CPE after the second passage. Although 18Ra-65 was not isolated from cells, the fiber, hexon, penton, and pol genes of 18Ra-64 were sequenced based on raccoon lung tissue, not virus isolated by cell culture. The negative results of the non-identification of CAdV-2 after the second passage of cells inoculated with 18Ra-65 are shown in Appendix A.

### 3.3. Sequence and Phylogenetic Tree Analysis

The entire genomic sequence of CAdV-2/18Ra-54 was obtained through PCR and 5’ and 3’ RACE and was subjected to a structure analysis. The genetic structure of CAdV-2/18Ra-54 was the same as that of previously identified CAdV-2 (Figure 2). The structure and location of each gene was the same between the two strains, and no differences in sequence were found.

The similarity of the raw sequences of each fiber, hexon, penton, and pol gene of the CAdV-2/18Ra-54 and CAdV-2/18Ra-65 strains obtained through sequencing is 99.9%, 100%, 99.9%, and 99.9%, respectively. As a result of a BLAST search, fiber, hexon, and penton genes showed a 99.86–100% similarity with that of the Toronto A26/61 strain (GenBank accession number U77082). The hexon gene showed a 100% similarity to that of the YCA-18 strain (GenBank accession number EF508034). The entire sequence of canine adenovirus 2 isolated in this study and the sequence data of each major gene were assigned an accession number in GenBank (OP644981, OP645070-OP645075). When comparing the amino acids of the CAdV-2/Toronto A26/61 and of the CAdV-2/18Ra-54 strains, the 315th amino acid valine (V) has been substituted with glutamine (Q) (Table 3). 

When comparing the amino acids of the CAdV-2/Toronto A26/61 and CAdV-2/18Ra-65 strains, the 315th amino acid proline (P) of the fiber gene and the 545th amino acid valine (V) of the penton base gene have been deleted. Glycine (G), the 891st amino acid of the hexon gene, has been substituted with V (Table 4).

CAdV-1 and CAdV-2 have different serotypes but are located close to each other in the phylogenetic tree. In addition, CAdV-1 and 2 isolated from foxes and wolves were closely related to those isolated from dogs (Figure 3).

The trees were produced using the NJ method and were bootstrapped with 1000 replicates. The branch lengths indicate the numbers of nucleotide substitutions per site. The sequences from this study are indicated in bold. Only bootstrap values at or above a 50% rate are shown.

## 4. Discussion

A study was conducted to evaluate whether canine adenovirus types 1 and 2, which are prevalent in domestic dogs, are likely to infect wild raccoon dogs as well.

The cell rounding and detachment phenomenon known as CPE of CAdV-2 was clearly observed in Vero cells inoculated with the 18Ra-54 strain obtained from lung tissue supernatant [25], whereas Vero cells inoculated with the 18Ra-65 strain from lung tissue supernatant did not manifest CPE. The strength of the band detected after PCR was weak, implying that the inoculation and proliferation of the intracellular virus were not performed well because of the low concentration of 18Ra-65 virus.

The CAdV-2/18Ra-54 strain isolated from wild raccoon dogs in Korea belongs to the same group as the Toronto A26/61 strain (Y77082), which is a CAdV-2 isolated in Canada. CAdV-2 in dogs, a representative companion animal in Korea, and CAdV-2 in wild raccoon dogs are closely related. The Toronto A26/61 strain is used in canine vaccines, and it is conceivable that viral contagious diseases such as the highly contagious canine adenovirus could be transmitted to other individuals through vaccinated pets. Additional research may be needed to identify issues relating to the safe use of vaccines to prevent outbreaks, wild animal transmission, and infection by vaccinated animals.

CAdV-1 and CAdV-2 were clearly distinguished from other *Mastadenoviruses* in the phylogenetic tree and showed high degrees of similarities with each serotype. Both CAdV-2/18Ra-54 and CAdV-2/18Ra-65 strains isolated from wild raccoon dogs were most genetically similar to the CAdV-2/Toronto A26/61 strain discovered in 1996 in dogs in Canada. According to a 2019 study, the CAdV-2 APQA1601 strain, isolated from Korean dogs, is also 99.9% identical to the Toronto A26/61 strain [26]. CAdV-2—prevalent in dogs in Korea—can infect wild raccoon dogs and is highly likely to cause an epidemic that can transfer from pet dogs to wild animals. In the future, constant surveillance will be required to monitor the spread of CADV-2 in the wild.

Although less common than other infectious diseases prevalent in wild raccoon dogs in Korea, the prevalence of CAdV-2, as ascertained through our study, was 1.9%, indicating that CAdV-2 infection does impact Korean wildlife. CAdV-2 exists in a variety of forms—from mild to severe—and transmission is known to occur readily through saliva and airborne droplets by behaviors such as biting and licking [27]. CAdV-1 was not detected in this study, but CAV-1 infection was reported in foxes and otters showing symptoms of anorexia and weight loss in another domestic study [28,29]. CAdV-2 was also detected in some dogs in Korea [30].

In conclusion, no new mutations of canine adenovirus were discovered in this study, and the most prevalent strain detected in Korean wild raccoon dogs was found to have similar genetic characteristics to the Toronto A24/61 strain isolated in Canada in 1976. This study is a multidisciplinary approach ranging from the investigation of the prevalence of canine adenovirus in wild raccoon dogs in Korea and virus detection to virus isolation and genetic characterization using cell culture and other molecular biological techniques. This study contributes to the understanding of canine adenovirus 2 in wild animals in Korea and suggests that there is a need for continuous adenovirus monitoring in the future.

## Figures and Tables

**Figure 1 vetsci-09-00591-f001:**
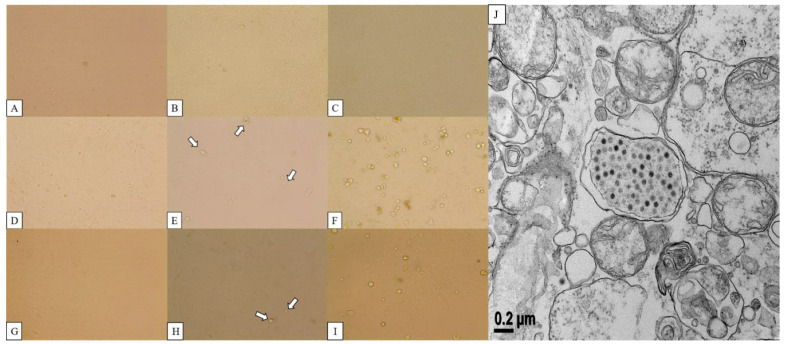
(**A**–**C**) Normal Vero cell; (**D**–**F**) CPE after inoculation of CAdV-2/18Ra-54; (**G**–**I**) CAdV-2/18Ra-65 in Vero cells and (**J**) the isolated CAdV-2/18Ra-54 observed with Bio-TEM.

**Figure 2 vetsci-09-00591-f002:**
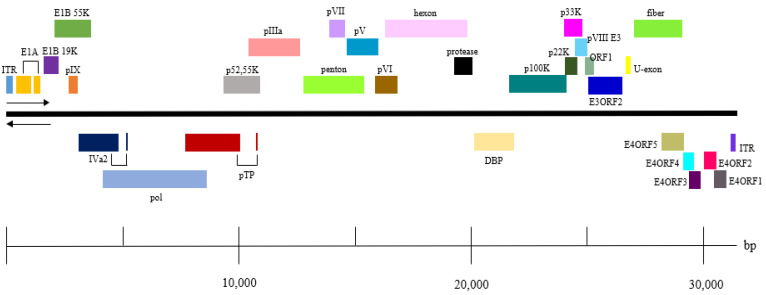
Genome structure of CAdV-2/18Ra-54.

**Figure 3 vetsci-09-00591-f003:**
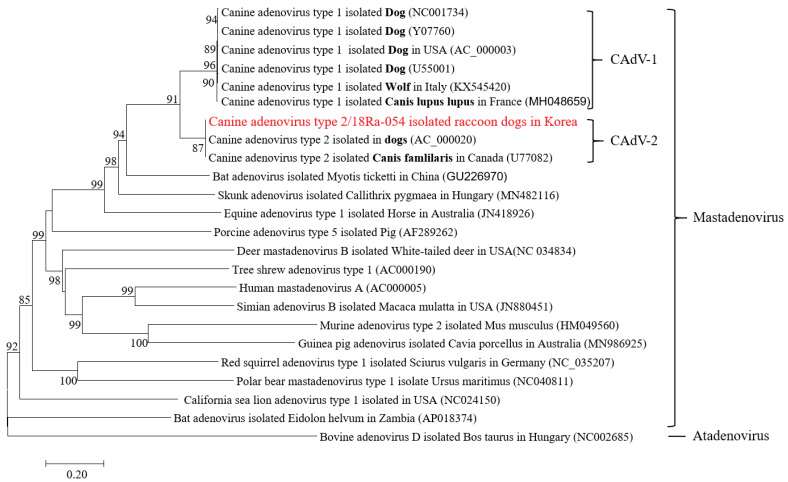
Phylogenetic tree of the entire nucleotide sequences of CAdV-2/18Ra-54.

**Table 1 vetsci-09-00591-t001:** Information on the region, season of discovery, and sex of the raccoon dog samples.

Groups	Subgroups	Number of Raccoon Dogs
Region	Chungcheong-do	12
	Gangwon-do	16
	Gyeonggi-do	31
	Gyeongsang-do	17
	Jeolla-do	28
	Seoul city	1
Season	Spring	7
	Summer	13
	Autumn	73
	Winter	8
	Unknown	8
Sex	Female	10
	Male	33
	Unknown	62

**Table 2 vetsci-09-00591-t002:** The number of positive results for presence of CAdV-1 and 2 in the total number of raccoon dogs and tissue samples tested.

Serotype	Number Positive/ Number of Raccoon Dogs	Number Positive/ Number of Tissues	Tissues			
			Liver	Kidney	Lung	Intestine
CAdV-1	0/105	0/414	0/105	0/105	0/104	0/100
CAdV-2	2/105	6/414	1/105	1/105	2/104	2/100

**Table 3 vetsci-09-00591-t003:** Comparison of amino acid sequence identity between Toronto A26/61 strain (GenBank accession number U77082) and CAdV-2/18Ra-54.

Gene	Strain	Size CAdV-2 (Amino Acid)	Amino Acid Sequence Identity (%)	Amino Acid Residues	
E2A	Toronto A26/61	454	-	315	V
	CAdV-2/18Ra-54	454	99.78		Q

**Table 4 vetsci-09-00591-t004:** Comparison of the size and amino acid sequence identity of the major genes (fiber, hexon, penton, and pol) between Toronto A26/61 (GenBank accession number U77082) and CAdV-2/18Ra-65 strains.

Gene	Strain	Size CAdV-2 (Amino Acid)	Amino Acid Sequence Identity (%)	Amino Acid Residues	
Fiber	Toronto A26/61	540	-	315	P
	CAdV-2/18Ra-65	539	99.94		-
Hexon	Toronto A26/61	904	-	891	G
	CAdV-2/18Ra-65	904	99.89		V
Penton	Toronto A26/61	476	-	454	V
	CAdV-2/18Ra-65	475	99.86		-
Pol	Toronto A26/61	1145	-	467	-
	CAdV-2/18Ra-65	1145	100		-

## Data Availability

Not applicable.

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
