# Peer review of "Genetic Characteristics of Canine Adenovirus Type 2 Detected in Wild Raccoon Dogs (Nyctereutes procyonoides) in Korea (2017–2020)"

_vetsci, 2022, doi:10.3390/vetsci9110591_

Round 1
Reviewer 1 Report
Major comment:
1. There is not enough background information or analysis to support the author's conclusion (speculation?) that the origin of the CAdV-2 found in this study is from the canine vaccine. There is overinterpretation of the results in line 18-20, 35-37 and 238-241. These statements need to be much better supported and explained, or removed.
Minor comments:
2. In line 146, the authors state that the strains of the two viruses were 18Ra-54 and 18Ra-65, but they don't explain how these strain names were assigned or determined.
3. On line 171, they refer to 18Ra-64. Is this a typo, or another strain?
4. There is no information on whether the sequences of the viruses that are described in this paper were submitted to GenBank. They should be submitted and the accession numbers listed.
5. In line 247-248, it seems that the authors are talking about prevalence of virus detected in serum, but the paper they cited is talking about seroprevalence (antibodies), which is different.
6. Supplemental figures S2 and S3 are not interpretable because of a lack of a clear description of what is in each lane of the gel. There are 12 lanes in each gel, but the authors only identify 6 of the samples.
7. Table 3 and lines 190-196 are very unclear about which viral proteins these amino acids are from. Present the data as in Table 4, and state which genes you are showing.
8. Examples of unclear writing or English errors that need to be fixed: Lines 22-23, 27, 33-34, 49-51, 190, 219
Author Response
Review 1
Major comment:
There is not enough background information or analysis to support the author's conclusion (speculation?) that the origin of the CAdV-2 found in this study is from the canine vaccine. There is overinterpretation of the results in line 18-20, 35-37 and 238-241. These statements need to be much better supported and explained, or removed.
RESPONSE: Our speculation that the origin of the CAdV-2 found in this study is a canine vaccine is low but plausible. All over-interpreted parts have been modified to raise only a slight possibility.
The conclusion of this study is to analyze the prevalence of CAdV-1 and CAdV-2 and the genetic characteristics of the prevalent CAdV-2 in raccoon dog, a wild animal in Korea.
Minor comments:
- In line 146, the authors state that the strains of the two viruses were 18Ra-54 and 18Ra-65, but they don't explain how these strain names were assigned or determined.
RESPONSE: We named the virus isolated by inoculating the cells as CAdV-2/18Ra-54, and modified by marking line 157.
- On line 171, they refer to 18Ra-64. Is this a typo, or another strain?
RESPONSE: On line 171, 18Ra-64 is correct, and we have corrected it to be clearer.
- There is no information on whether the sequences of the viruses that are described in this paper were submitted to GenBank. They should be submitted and the accession numbers listed.
RESPONSE: We have registered the entire gene of 18Ra-54 and the major genes of 18Ra-54 and 18Ra-65 with GenBank and have been given accession numbers OP644981 and OP645070-OP645075.
- In line 247-248, it seems that the authors are talking about prevalence of virus detected in serum, but the paper they cited is talking about seroprevalence (antibodies), which is different.
RESPONSE: We corrected and removed the relevant content, and the cited papers were excluded from the reference.
- Supplemental figures S2 and S3 are not interpretable because of a lack of a clear description of what is in each lane of the gel. There are 12 lanes in each gel, but the authors only identify 6 of the samples.
RESPONSE: We have corrected and supplemented as follows: Supplementary Figures S2 and S3 show the sequence from lane 1 to passage 1 of the infected cell culture medium (medium), passage 1 cell lysate, passage 2 culture, and passage 2 cell lysate. Raw lane 11 is passage 6 medium, lane 12 is passage 6 cell lysate.
- Table 3 and lines 190-196 are very unclear about which viral proteins these amino acids are from. Present the data as in Table 4, and state which genes you are showing.
RESPONSE: We have rearranged and corrected the base sequence and amino acid sequence of Table 3 in a form similar to that of Table 4 format.
- Examples of unclear writing or English errors that need to be fixed: Lines 22-23, 27, 33-34, 49-51, 190, 219
RESPONSE: We corrected the English error in the part you pointed out: lines 22-23, 27, 33-34, 49-51, 190, 219
Reviewer 2 Report
The manuscript presents the sequence analysis of the two field isolates of canine adenovirus 2. The work would be much more interesting if an in vivo analysis of the properties of these strains is included, considering their striking difference in the ability to infect cultures of Vero cells, despite nearly identical nucleotide sequences. Based on the detected mutations, can Authors speculate what causes different behavior of these strains?
I would recommend testing the replication of these strains in other cell lines.
Some easy experiments on the replication in Vero cells could be performed, to pinpoint the difference between strains. Is strain 18Ra-65 infecting the cells? Is the DNA of 18Ra-65 replicating? If a suitable antibody is available, a western blot analysis could be made.
The genomes of both strains have to be deposited in Gene Bank, with Accession No. provided in the manuscript
Why only 2/100 samples tested in this study were CAdV-2 positive, if the previous research, (cited as reference 16) found a prevalence over 50%?
The first paragraph of the introduction is hard to understand for the reader unfamiliar with the Adenovirus structure. Reference to figure 2 might help, but the whole paragraph should be improved.
I found several minor issues in the manuscript:
L. 27 Only CAdV-2 was identified was detected in two raccoon dogs, - Delete word identified
L.29Adenovirus were successfully isolated from PCR positive samples – was, not were
L 162 and CAdV-2/18Ra-54 was confirmed to be found in cells from all passages inoculated with strain 18Ra-54 in the lung tissue. The confusing phrase, suggests that cells were in lung tissue. Replace was with were.
Author Response
Review 2
The manuscript presents the sequence analysis of the two field isolates of canine adenovirus 2. The work would be much more interesting if an in vivo analysis of the properties of these strains is included, considering their striking difference in the ability to infect cultures of Vero cells, despite nearly identical nucleotide sequences. Based on the detected mutations, can Authors speculate what causes different behavior of these strains?
I would recommend testing the replication of these strains in other cell lines.
Some easy experiments on the replication in Vero cells could be performed, to pinpoint the difference between strains. Is strain 18Ra-65 infecting the cells? Is the DNA of 18Ra-65 replicating? If a suitable antibody is available, a western blot analysis could be made.
The genomes of both strains have to be deposited in Gene Bank, with Accession No. provided in the manuscript
Why only 2/100 samples tested in this study were CAdV-2 positive, if the previous research, (cited as reference 16) found a prevalence over 50%?
The first paragraph of the introduction is hard to understand for the reader unfamiliar with the Adenovirus structure. Reference to figure 2 might help, but the whole paragraph should be improved.
RESPONSE: Even though the same amount of RNA was used, the PCR band strength of all 18Ra-54 tissue samples was much stronger than that of 18Ra-65. Also, the symptoms of CPE seen in cells were more severe in 18Ra-54. We attributed this difference to the stronger viral infectivity and concentration of 18Ra-54 than 18Ra-65. For the above reasons, 18Ra-54 succeeded in isolating the virus and obtained the entire gene, but 18Ra-65 failed to isolate the virus. The gene sequence of the major genes of 18Ra-65 was obtained from the tissue sample used for CAdV-2 screening.
We have registered the entire gene of 18Ra-54 and the major genes of 18Ra-54 and 18Ra-65 with GenBank and have been given accession numbers OP644981 and OP645070-OP645075.
In the previous study (cited as reference 16), the prevalence of over 50% was due to a serological study. Since it is a different point of view from this study, it has been corrected and deleted and excluded from the references.
And for easy understanding, the first paragraph of the introduction has been abbreviated and briefly explained.
I found several minor issues in the manuscript:
- 27 Only CAdV-2 was identified was detected in two raccoon dogs, - Delete word identified
RESPONSE: We corrected in line 27.
L.29Adenovirus were successfully isolated from PCR positive samples – was, not were
RESPONSE: We corrected in line 29.
L 162 and CAdV-2/18Ra-54 was confirmed to be found in cells from all passages inoculated with strain 18Ra-54 in the lung tissue. The confusing phrase, suggests that cells were in lung tissue. Replace was with were.
RESPONSE: We corrected in line 161.
Round 2
Reviewer 2 Report
Thank you for addressing most of my concerns. Regarding the issue of replication in Vero cells, I do understand that one strain is more infectious than the other. I was asking if it is possible to explain this difference in the behavior of the strains based on the differences in their DNA sequences. I accept that it may not be possible based on the available information.
There are some language issues remaining:
L.18 - "Therefore, it contributes to the need for further studies" , what contributes to the need? It would be better to say: Further studies are needed to...
L.173 - phrase should be re-worded (how are virus genes successful or not successful in virus isolation?)
Author Response
Dear Dr. Patrick Butaye
We appreciate the opportunity to submit to the Ministry of Veterinary Science a draft revised manuscript titled "Genetic Characteristics of Canine adenovirus type 2 Detected Wild Raccoon dogs (Nyctereutes procyonoides) in Korea, 2017-2020". We also appreciate the time and effort you and each reviewer devoted to providing insightful feedback on how to enhance our paper. Therefore, we are pleased to resubmit the article for further consideration. We've incorporated changes that reflect the detailed suggestions you have kindly provided. We also hope that the fixes and responses provided below satisfactorily address any issues and concerns pointed out by you and our reviewers.
To facilitate your review of our revisions, the following is a point-by-point response to the questions and comments.
Reviewer 2
Comments and Suggestions for Authors
Thank you for addressing most of my concerns. Regarding the issue of replication in Vero cells, I do understand that one strain is more infectious than the other. I was asking if it is possible to explain this difference in the behavior of the strains based on the differences in their DNA sequences. I accept that it may not be possible based on the available information.
RESPONSE: As suggested, it is thought that the difference in replication in Vero cells may occur due to the difference in DNA sequence between the two strains in this study. However, the entire gene sequence of the 18Ra-65 strain could not be obtained, and the sequence differences of some major genes (fiber, hexon, penton and pol genes) are thought to be insufficient to explain all the differences in replication in Vero cells. Based on the points you pointed out I think there is a need for further research in the future.
There are some language issues remaining:
L.18 - "Therefore, it contributes to the need for further studies" , what contributes to the need? It would be better to say: Further studies are needed to...
RESPONSE: We corrected in line 18.
L.173 - phrase should be re-worded (how are virus genes successful or not successful in virus isolation?)
RESPONSE: We corrected in line 173.